# IgA Anti-β2-Glycoprotein I Antibodies as Markers of Thrombosis and Severity in COVID-19 Patients

**DOI:** 10.3390/v16071071

**Published:** 2024-07-03

**Authors:** Susana Mellor-Pita, Pablo Tutor-Ureta, Paula Velasco, Aresio Plaza, Itziar Diego, José Vázquez-Comendador, Ana Paula Vionnet, Pedro Durán-del Campo, Víctor Moreno-Torres, Juan Antonio Vargas, Raquel Castejon

**Affiliations:** 1Systemic Autoimmune Diseases Unit, Department of Internal Medicine, IDIPHIM (Puerta de Hierro University Hospital Research Institute), Hospital Universitario Puerta de Hierro Majadahonda, 28222 Madrid, Spain; susanamellor@hotmail.com (S.M.-P.); pablo.tutor@hotmail.com (P.T.-U.); pvelascog@gmail.com (P.V.); i.diego.yague@gmail.com (I.D.); jvcomendador@salud.madrid.org (J.V.-C.); pedrodurandc@hotmail.com (P.D.-d.C.); victor.moreno.torres.1988@gmail.com (V.M.-T.); juanantonio.vargas@salud.madrid.org (J.A.V.); 2Department of Medicine, Facultad de Medicina, Universidad Autónoma de Madrid, 28029 Madrid, Spain; 3Department of Immunology, Hospital Universitario Puerta de Hierro Majadahonda, 28222 Madrid, Spain; aresio.plaza@salud.madrid.org (A.P.); anapaula.vionnet@salud.madrid.org (A.P.V.)

**Keywords:** antiphospholipid antibodies, COVID-19, thrombosis, severity

## Abstract

Patients with COVID-19 may develop a hypercoagulable state due to tissue and endothelial injury, produced by an unbalanced immune response. Therefore, an increased number of thromboembolic events has been reported in these patients. The aim of this study is to investigate the presence of antiphospholipid antibodies (aPL) in COVID-19 patients, their role in the development of thrombosis and their relationship with the severity of the disease. In this retrospective study, serum samples from 159 COVID-19 patients and 80 healthy donors were analysed for the presence of aPL. A total of 29 patients (18.2%) and 14 healthy donors (17.5%) were positive for aPL. Nineteen COVID-19 patients (12%) but no healthy donor presented a positive percentage of the IgA isotype aPL. IgA anti-β2-glycoprotein I antibodies (anti-β2GPI) were the most frequent type (6.3%) in patients but was not detected in any healthy donor. The positivity of this antibody was found to be significantly elevated in patients with thromboembolic events (25% vs. 5%, *p* = 0.029); in fact, patients with positive IgA anti-β2GPI had an incidence of thrombosis over six times higher than those who had normal antibody concentrations [OR (CI 95%) of 6.67 (1.5–30.2), *p* = 0.014]. Additionally, patients with moderate-severe disease presented a higher aPL positivity than patients with mild disease according to the Brescia (*p* = 0.029) and CURB-65 (*p* = 0.011) severity scales. A multivariate analysis showed that positivity for IgA anti-β2GPI is significantly associated with disease severity measured by CURB-65 [OR (CI 95%) 17.8 (1.7–187), *p* = 0.0016]. In conclusion, COVID-19 patients have a significantly higher positive percentage of the IgA isotype aPL than healthy donors. IgA anti-β2GPI antibodies were the most frequently detected aPL in COVID-19 patients and were associated with thrombosis and severe COVID-19 and are thus proposed as a possible marker to identify high-risk patients.

## 1. Introduction

Thrombosis is a severe complication of COVID-19. An increased risk of venous and arterial thromboembolic events such as deep vein thrombosis, pulmonary embolism, strokes and myocardial infarctions has been described [1].

SARS-CoV-2 enters the host cells by binding the SARS-CoV-2 spike to the angiotensin-converting enzyme 2 (ACE2) receptors, abundant on type II alveolar epithelial cells, causing direct virus-mediated tissue damage followed by an activation of the innate immune system which releases cytokines [2]. In addition, endothelial cells express ACE2 receptors allowing infection by SARS-CoV-2. These direct viral effects as well as perivascular inflammation may contribute to endothelial injury (endothelialitis) [3]. Patients may develop a hypercoagulable state due to this tissue and endothelial injury produced by an unbalanced immune response.

Several studies based on autopsies of deceased COVID-19 patients showed a greater degree of endothelialitis, microangiopathy and thrombosis in their lungs, as well as higher tissue expression of IL-6 and TNFα compared to that found in the lungs of patients who died from acute respiratory distress syndrome secondary to influenza A1 (H1N1) infection and uninfected control lungs [3,4]. The evidence from many COVID-19 studies points to endothelial damage as a key component in the progression of the disease to its later complicated stages. Endothelial damage is associated with the loss of the anticoagulant properties of the endothelium, which may contribute to the hypercoagulable state of these patients as well as an overactivation of the complement cascade in SARS-CoV-2, which in turn can also promote acute and chronic inflammation, intravascular coagulation and endothelial cell injury [5]. The endothelial damage caused by COVID-19 is therefore at the crossroads of the hypercoagulable state, impaired fibrinolysis, activation of the complement system and the degradation of the glycocalyx layer, all of which are processes linked in the pathogenesis of COVID-19 complications.

Antiphospholipid syndrome (APS) is a frequent cause of acquired thrombophilia that promotes thrombosis in arterial and venous vessels of all sizes and gestational morbidity in patients with persistently high levels of antiphospholipid antibodies (aPL). The aPL included in the classification criteria are lupus anticoagulant, anticardiolipin (anti-CL) and anti-beta2glycoprotein I antibodies (anti-β2GPI) of the IgG and IgM isotypes. There are also extra-criteria aPL that have been associated with APS which are not included in the classification criteria such as anti-CL and anti-β2GPI of the IgA isotype and anti-phosphatidylserine/prothrombin (anti-PS/anti-PT) [6].

Viral infections are well-known triggers of antiphospholipid antibody production via molecular mimicry in certain predisposed individuals, anti-CL being the most commonly reported. Most of these virus-associated aPL are thought to be transient. They may represent an epiphenomenon, but in some cases, an increased risk of thromboembolic events has been described [7,8,9].

The mechanisms of thrombotic events in COVID-19 patients are not fully known. There seems to be molecular/cellular pathways that involve a dysregulated renin–angiotensin–aldosterone system and excessive innate immune response to SARS-CoV-2, which may lead to thrombosis [1]. On the other hand, coagulation test abnormalities have been observed and are most likely a result of a profound inflammatory response. An increase in D-dimer and fibrinogen has been described in these patients, as well an increase in coagulation times, activated partial thromboplastin time (APTT), and prothrombin time (PT). The elevation of these values seems to be related to an increased risk of acute respiratory distress syndrome (ARDS), intensive care unit (ICU) admissions and disease severity as well as a higher mortality [10].

The identification of the risk of progression to severe disease is crucial in preventing respiratory failure and mortality in COVID-19 patients. The elevated C-reactive protein, D-dimer, lactate dehydrogenase and IL-6 as well as severe lymphopenia have been reported to be associated significantly with worse outcomes [11]. Recently, Kang DH et al. have explored the prognostic value of quantitative high-resolution computed tomography (QCT) lung COVID scores along with laboratory inflammation markers and found that patients with a high mixed-disease pattern score (≥10%) were likely to experience rapid progression within seven days, suggesting that QCT COVID scores at admission could predict rapid progression in COVID-19 patients [12].

A high prevalence of aPL in patients with COVID-19 has been reported in some studies [13,14]. However, the results show discrepancies in the data on the prevalence of aPL and their role in the pathogenesis of thrombosis in these patients [15,16].

The aim and the novelty of this study are to determine, in a larger group of patients and healthy donors compared to previous studies, the presence of antiphospholipid antibodies in COVID-19 patients and to assess their association with thrombosis and the severity of the disease and, thus, whether they can be used as possible patient risk markers and as a guide to their treatment

## 2. Patients and Methods

This retrospective observational cohort study was performed at the Hospital Universitario Puerta de Hierro, Majadahonda, Madrid, Spain. The study population included 159 adult patients (≥18 years) with COVID-19, confirmed by PCR on nasopharyngeal swab, who were admitted consecutively between 24 and 31 March 2020. The control group consisted of 80 healthy blood donors matched with the patients by age and gender.

In these patients, the presence of antiphospholipid antibodies (specifically anticardiolipin, anti-β2-glycoprotein I, anti-phosphatidylserine and anti-prothrombin antibodies) in serum samples was determined shortly after admission (from 24 to 48 h in all cases). Demographic and clinical data were obtained from electronic medical records. The patients were followed for the duration of their hospital stay, for a maximum of 3 months.

The study was approved by the Research Ethics Commission of the Puerta de Hierro University Hospital (reference PI 94/20) and oral consent to register their clinical information into a database and further use of biological material was requested from patients.

At the time of the study the following baseline characteristics were assessed:Demographic variables: cardiovascular risk factors, cardiovascular disease, thrombophilia, malignancy, connective tissue disease, lung or kidney diseases, pregnancy, or puerperium and treatment before admission;Clinical features: fever, cough, dyspnoea, diarrhoea, anosmia and ageusia;Routine laboratory tests: complete blood cell count, coagulation test, D-dimer, creatinine, lactate dehydrogenase, C-reactive protein, ferritin and interleukin-6 levels;Severity scales: Brescia-COVID Respiratory Severity Scale, CURB-65 and neutrophil-to-lymphocyte ratio (NLR);PaO_2_/FiO_2_ (arterial oxygen partial pressure (mmHg) to fractional inspired oxygen ratio) and SO_2_/FiO_2_ (oxygen saturation to fractional inspired oxygen ratio);Percentage of chest affected according to X-ray;Complications during hospital stay: deep vein thrombosis (DVT), pulmonary embolism (PE), other thromboses and death;Treatment during hospital stay: corticosteroids, corticosteroid pulses, anakinra, tocilizumab and low-molecular-weight heparin (LMWH);Admission to ICU.

### 2.1. Antiphospholipid Antibodies Quantification

Quantification of antiphospholipid antibodies (aPL) was carried out in serum samples using the enzyme immunoassay (ELISA) technique. For our analysis, we used the manufacturer’s (Grifols, Barcelona, Spain) suggested thresholds to define the antibody concentration considered as negative, with higher values considered positive. The negative values for anticardiolipin antibodies (anti-CL) (IgG, IgM and IgA) and for anti-β2-glycoprotein I (anti-β2GPI) (IgG, IgM and IgA) were ≤12 IU/mL. The determination of anti-phosphatidylserine (anti-PS) and anti-prothrombin (anti-PT) was performed in a combined ELISA of both antigens. The positive samples from this determination were analysed in another ELISA specific for anti-PT that then allowed the identification of positivity for both anti-PS and anti-PT. The negative value for anti-PS/anti-PT (IgG and IgM) was ≤16 IU/mL and ≤12 IU/mL for anti-PS/anti-PT (IgA).

The lupus anticoagulant assay was not carried out because false-positive results, induced by the acute inflammatory condition of patients, can be obtained.

### 2.2. Statistical Analysis

Categorical variables were reported as frequencies and percentages while continuous variables were presented by their mean and standard deviation. The Kolmogorov–Smirnov test was used to evaluate data distribution. The correlation between the aPL concentration and clinical variables was analysed using the Spearman test. The significance of the differences and outcomes of the presence of antiphospholipid antibodies on thrombosis and disease severity between aPL positive patients and patients with non-positive aPL were determined by the Chi-square test, Fisher’s test or Student’s *t*-test, as appropriate. A multivariable logistic regression analysis was performed to determine the association of antibodies with the severity of the disease. The data were stored in Excel 2016 and processed using the statistical program SPSSv25. For all the analyses, a significance level of 0.05 was set.

## 3. Results

A total of 159 patients with microbiological diagnosis of COVID-19 were included, and the data collected from these patients are shown in Table 1. Healthy controls were 75% men with a mean age of 55.9 ± 5.2 years.

### 3.1. Presence of Antiphospholipid Antibodies

A total of 29 out of 159 patients (18.2%) and 14 healthy donors (17.5%) were positive for any type of aPL. The presence and concentration of each specific antibody are shown in Table 2.

Interestingly, nineteen COVID-19 patients (12%) presented a positive percentage of the IgA isotype aPL, but no healthy donor had IgA isotypes. IgA anti-β2GPI was the most frequently detected, being present in 10 (6.3%) of the COVID-19 patients.

Anti-CL were detected in 3 healthy donors (3.5%) and in 12 patients (7.6%), the IgG isoform being the most frequent with 6 patients (3.8%); 1 patient was positive for IgG/IgA anti-CL.

In the combined anti-PS/anti-PT test, a positive result was obtained in 12.5% of the control group and 8.2% of patients, with six (3.8%) of the samples positive for IgG/IgM, six samples positive for IgA and another one (0.6%) positive for IgG/IgM/IgA. A total of eight (5.0%) patients presented any type of anti-PT elevation, three (1.89%) were positive for IgG/IgM anti-PT, one (0.6%) was positive for IgM, three were positive for IgA anti-PT and one was positive for IgG/IgM/IgA anti-PT antibodies), five (3.1%) presented any type of anti-PS elevation, two positive samples (1.3%) corresponded to IgG/IgM anti-PS and three (1.9%) were positive for IgA anti-PS.

Combined positivity for anti-CL and anti-PS/anti-PT was observed in two patients, one patient was positive for anti-CL and anti-β2GPI, and another one was positive for anti-β2GPI and anti-PS/anti-PT. Additionally, two patients showed positive levels of the three subtypes of antiphospholipid antibodies analysed.

No differences were found between the patients with positive and negative aPL and demographic data (age and gender), cardiovascular risk factors and comorbidities.

All tests were carried out within 11.5 ± 4.5 days of the onset of COVID-19 symptoms. The 29 patients with positive aPL had experienced significantly more days of COVID-19 symptoms than the patients with negative aPL (13.6 ± 5.0 vs. 11.1 ± 4.2, *p* = 0.003).

During their hospital stay, 84.2% of the patients were treated with low-molecular-weight heparin (LMWH), 74.8% with corticosteroids and 39.8% with tocilizumab. No differences were found between the presence of antiphospholipid antibodies and the treatment the patients received.

### 3.2. Antiphospholipid Antibodies, Thrombosis and Coagulation Test in Patients with COVID-19

During the hospital stay of the patients included in this study, the occurrence of venous and arterial thrombosis was recorded. Twelve patients (7.6%) were diagnosed with thrombosis: three patients (1.9%) presented DVT, five patients (3.1%) PE, two patients (1.3%) both DVT and PE and two patients (1.3%) another type of thrombosis (acute myocardial infarction, stroke). Three of the patients with thrombosis were positive for any type of aPL, all were positive for IgA anti-β2GPI and one was also positive for IgG/IgM anti-PT.

An analysis of the relationship between the presence of aPL and the incidence of thrombosis showed an OR (CI 95%) of 6.67 (1.5–30.2), *p* = 0.014, meaning that patients with positive IgA anti-β2GPI have an incidence of thrombosis over six times higher than those who have normal antibody concentrations; the patients with thrombosis had a significantly higher positive percentage of IgA anti-β2GPI than the group of patients without thrombosis (25.0% vs. 4.8%, *p* = 0.029) (Figure 1).

No significant differences were found in the presence of aPL and the type of thrombotic event experienced.

The relationship between the presence of thrombosis and the serum aPL concentration was also analysed. Although not statistically significant, the values of IgA anti-β2GPI were markedly elevated in patients with thrombosis compared to patients without thrombosis (30.6 ± 82.3 IU/mL vs. 8.5 ± 40.9 IU/mL) (Figure 1).

It should be noted that statistically significantly higher D-dimer values were found in patients with any positive aPL compared with negative aPL patients (6.3 ± 18.3 μg/mL vs. 3.6 ± 7.4 μg/mL, *p* = 0.029) and also in the group of patients specifically with positive IgA anti-β2GPI (12.9 ± 30.8 μg/mL vs. 3.5 ± 7.0 μg/mL; *p* = 0.017). As expected, patients with thrombosis presented significantly higher D-dimer levels than patients without thromboembolic events (24.8 ± 28.5 μg/mL vs. 2.4 ± 3.8 μg/mL; *p* < 0.001).

### 3.3. Antiphopholipid Antibodies and Severity of COVID-19

The relationship between aPL and the severity of infection produced by SARS-CoV-2, as indicated by the severity scales (Brescia, CURB-65 and NLR), was investigated.

On the Brescia scale, 96 (60.4%) mild-grade, 32 (20.1%) moderate-grade and 31 (19.5%) severe-grade patients were found. The percentage of patients with any positive aPL was significantly higher (*p* = 0.029), on the Brescia scale (Figure 2), in moderate-grade patients than in the mild- or severe-grade patients. Considering each aPL subtype’s concentration, significant differences (*p* = 0.002) were only found for IgA anti-β2GPI; the higher the concentration of this antibody, the greater the severity of the disease (4.9 ± 29.7 IU/mL for mild-grade, 17.8 ± 60.0 IU/mL for moderate-grade and 18.4 ± 63.8 IU/mL for severe-grade disease) (Figure 3).

On the CURB-65 scale, 91 patients (57.2%) presented low-risk pneumonia, 47 patients (29.6%) moderate-risk pneumonia and 21 patients (13.2%) high-risk pneumonia. Our results showed that there was a significantly higher number of patients with positive aPL who presented moderate and severe risk versus low-risk patients; this difference being statistically significant (*p* = 0.011) (Figure 2). Examining the concentrations of the different antibody subtypes, it was observed that the greater the severity of pneumonia, according to the CURB-65 scale, the more elevated the concentration of all types of IgA aPL, the statistical significances being *p* = 0.026 for IgA anti-CL, *p* = 0.005 for IgA anti-β2GPI and *p* = 0.009 for IgA anti-PS/anti-PT.

The NLR (mean ± SD) in our patients was 17.6 ± 58.8, with 134 patients (84.3%) presenting an NLR ≥ 3. Upon analysing the group of patients presenting aPL, no significant differences were found in the NLR (Figure 2). However, higher concentrations of IgA isotypes were present when the NLR was greater than three, with a statistical significance of *p* = 0.008 for IgA anti-CL, *p* = 0.001 for IgA anti-β2GPI and *p* = 0.018 for IgA anti-PS/anti-PT (Figure 3).

A multivariate analysis showed that positivity for IgA anti-β2GPI is significantly associated with disease severity measured by CURB-65 OR (CI 95%) 17.8 (1.7–187), where *p* = 0.0016.

Finally, the relationship between aPL and the incidence of ICU admission, death and readmissions was studied. There were 11 ICU admissions (6.9%), 11 deaths (6.9%) which do not correspond to the patients admitted to the ICU and 14 readmissions (8.8%). No significant differences were found between either the presence of aPL or the serum concentration of any antibody subtype and the severity outcomes.

## 4. Discussion

In this study, the presence of aPL in COVID-19 patients, as well as its association with the development of thrombosis and with the severity of the disease, was investigated. Our novel findings are that COVID-19 patients have a significantly higher positive percentage of IgA isotype antiphospholipid antibodies than healthy donors. As IgA anti-β2GPI was the most frequently detected antibody and was associated with thrombosis and severe COVID-19; it is proposed that this should be used as a marker for disease severity. Thus, it is suggested that COVID-19 patients should be screened for IgA anti-β2GPI, and those found to have higher titres should be subjected to close follow-up to prevent the development of thrombosis.

A similar prevalence of aPL was found in our COVID-19 patients (18.2%) compared to our healthy donor group (17.5%). In other studies, the prevalence of aPL with COVID-19 ranged from 1% to 95% [13,14,15,16,17,18,19]. Taha et al. conducted a meta-analysis and found that a pooled prevalence rate of one or more aPL was 46.8% (95% CI 36.1% to 57.8%) [20].

This suggests that SARS-CoV-2 is a trigger for aPL production similar to that observed in other viral infections. A likely mechanism is molecular mimicry. In COVID-19, this molecular mimicry mechanism may be mediated by pulmonary surfactants as they contain an abundance of phospholipid-binding proteins. Indeed Kanduc D. et al. discovered that almost 50% of the immunoreactive epitopes on the spike glycoprotein of SARS-CoV-2 share pentapeptides on human surfactant-related proteins [21]. The similarity between peptides in SARS-CoV-2 and surfactant proteins may trigger aPL.

In our study, no association between the age or the gender and the presence of aPL was found. Similar findings have previously been reported for both COVID-19 patients and the general population [22,23].

Although a widespread presence of aPL has been reported in patients with COVID-19, discrepancies in the data on the degree of prevalence and how it affects the pathogenesis of thrombosis exist. This may be a result of the diversity of the study population, the disease severity, the point in time during the disease when aPL were determined, the aPL type measured, and the detection procedure.

IgA anti-β2GPI was the most frequently detected antibody in the COVID-19 patients of our study (6.3%) and presented with moderate and high titres. This was also observed by Xiao et al. [24], Melayah et al. [25] and Serrano et al. [26] who reported IgA anti-β2GPI as the most frequently occurring aPL in COVID-19 patients (28.8%, 16.9% and 14.9%, respectively). This finding suggests that COVID-19 induced the IgA isotype aPL. Other authors reported an IgA anti-β2GPI prevalence of between 6.6% and 12% in critically ill patients [14,15,17].

Ig A is an isotype specialized in mucosal immunity, and SARS-CoV-2 makes its entry through the pulmonary and intestinal mucosa; the preferential production of the IgA isotype may be associated with the breakage of mucosal immune tolerance. Some studies have found a marked elevation in IgA antibodies against SARS-CoV-2 that is significantly associated with severe COVID-19 [27]. These data support that a strong IgA-driven immune response possibly emerges from bronchial-associated lymphoid tissue when SARS-CoV-2 affects the deeper respiratory system.

The IgM and IgG isotypes of aCL are the most commonly reported aPL associated with infection [8]; their prevalence in COVID-19 patients varies from 3% to 28% [13,15,16,18,28,29]. The prevalence of these antibodies in our study is 7.5%.

There are few published studies for anti-PS and anti-PT that incorporate their measurements along with other aPL, although one study of COVID-19 patients revealed IgG anti-PS/anti-PT as the most abundant [28]. However, in our analysis, these antibodies appear underrepresented.

Lupus anticoagulant is the most frequently detected aPL in COVID-19 patients, ranging in prevalence from 22.2% to 95% in different cohorts in medical wards and/or in the ICU, the highest prevalence being in critically ill COVID-19 patients [14,15,16,18,19]. Taha et al. found a pooled prevalence rate of 50.7% (95% CI 34.8% to 66.5%) [20]. Lupus anticoagulant assay outcomes can be influenced by several factors such as high levels of C-reactive protein and fibrinogen during an acute inflammatory condition which may induce false-positive results [15,30,31]. This may well apply to patients with COVID-19 and so may also induce false-positive results.

Another important issue is the point in time during the disease when aPL are determined. Xiao et al. reported that aPL emerged around a median time of 39 days post-disease onset; these findings show that aPL emerge at a later time-point, suggesting critically ill patients with longer disease durations are likely to have aPL [24]. In our study, the patients with positive aPL had experienced significantly more days of COVID-19 symptoms than the patients with negative aPL.

IgA anti-β2GPI is associated with thrombosis and severe COVID-19. The relationship between aPL and the occurrence of thrombosis is well documented; however, it is not certain whether their elevation can always lead to the appearance of TEE in COVID-19 patients. Despite the small number of patients in our study who presented thrombosis and positive aPL, the relationship between thrombosis and anti-β2GPI IgA was significant.

The incidence of APS-related events in asymptomatic IgA Anti-B2GP1 carriers was reported to be 3.1% per year [32]. There are studies that suggest that IgA anti-β2GPI is associated with thrombosis in patients with lupus (OR 2.8, 95% CI 1.3 to 6.2) [33], as well as a higher risk of acute myocardial infarction and stroke [34]. Indeed, since 2010, the task force of the Galveston International Congress on APS recommends testing for the IgA isotype (anti-CL and anti-β2GPI) in patients with clinical criteria of APS and with persistently negative aPL included in the classification criteria [35]. In recent years, several authors have evaluated the role of this antibody in the pathogenesis of APS, even proposing its inclusion in the classification criteria.

In the literature, there are divergent results on the presence of aPL and the risk of thrombotic events in COVID-19 patients. Xiao et al. found that patients with multiple aPL including IgA and IgG anti-β2GPI and IgA and IgG anti-CL displayed a significantly higher incidence of cerebral infarction compared to patients who were negative for aPL (*p* = 0.023) [24], while Amezcua-Guerra et al. suggested an association with pulmonary thromboembolism [29]. Le Joncour et al. showed a meaningful association of anticardiolipin and IgA anti-β2GPI with the occurrence of thrombotic events [36], and Gatto et al. found that patients with at least one aPL positive test show a trend in developing thrombosis compared with patients who were aPL-negative [13].

However, other published studies seem to rule out the relationship between aPL and thrombosis in COVID-19 patients. This may be because of the low number of patients included in each study; nevertheless, our study is one of the studies with the highest number of patients. In addition, most of these studies carried out lupus anticoagulant assays, and only a few of them analysed IgA anti-β2GPI; it should also be borne in mind that all these patients, as well as those included in our study, received prophylactic treatment with LMWH, which probably results in a lower occurrence of TEE [13,15,16,17,20].

As the pathogenesis of COVID-19 is related to an exaggerated inflammatory and immunological response, a higher concentration of aPL with disease severity might thus be expected. In our study, the results showed that the more critically ill patients had greater positivity for any type of aPL as well as a significant increase in IgA isotype aPL concentration. This was also found by Hasan Ali et al., who compared two cohorts with mild and severe COVID-19 and observed significantly elevated IgA anti-CL and IgA anti-β2GPI in the severe COVID-19 cohort [22], and by Garcia-Arellano et al., who found a significantly higher prevalence of IgA anti-β2GPI in moderate to severe COVID-19 patients [37]. In other published studies, those that registered the highest prevalence of aPL were those that selected patients with severe COVID-19 or patients admitted to the ICU [15,16,20,24,38]. This can be partially explained by the extensive inflammation, cellular damage and apoptosis in critically ill patients that can induce aPL production. These findings suggest that the presence of aPL in COVID-19 patients, especially the IgA isotype, could be used as markers for COVID-19 severity.

The main limitations of this research are that it is a single-centre retrospective study, it does not include longitudinal measurements of antibodies and the follow-up of the patients is short-term. Positive results of anti-β2GPI, anti-CL, anti-PS and anti-PT need to be confirmed a second time, after twelve weeks, to confirm persistent positivity. We do not know if the concentration of aPL studied fades with the resolution of COVID-19. Nevertheless, the inclusion of the three serotypes of anti-β2GPI, anti-CL, anti-PT and anti-PS and the size of the sample are the strengths of the study.

The clinical implications of our findings are that aPL should be screened in COVID-19 patients as markers of severity, a close follow-up of these patients for the development of thrombosis should be undertaken, and high-dose prophylactic anticoagulation should be used.

## 5. Conclusions

IgA anti-β2GPI was the most frequent aPL detected in the serum of COVID-19 patients included in this study and was associated with thrombosis and severe COVID-19. Further prospective studies are needed with other patient cohorts, in addition to increased sample size and long-term follow-up to ascertain the usefulness of these markers in identifying patients with high risk of thrombosis and severe COVID-19.

## Figures and Tables

**Figure 1 viruses-16-01071-f001:**
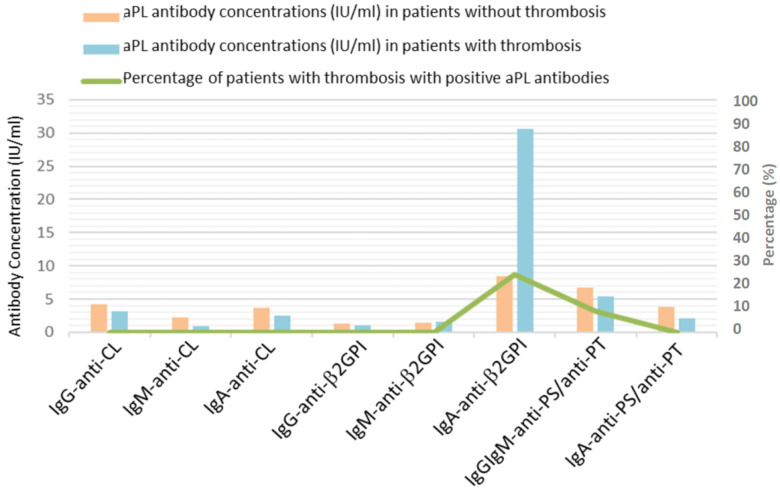
Presence of antiphospholipid antibodies and the incidence of thrombosis in COVID-19 patients.

**Figure 2 viruses-16-01071-f002:**
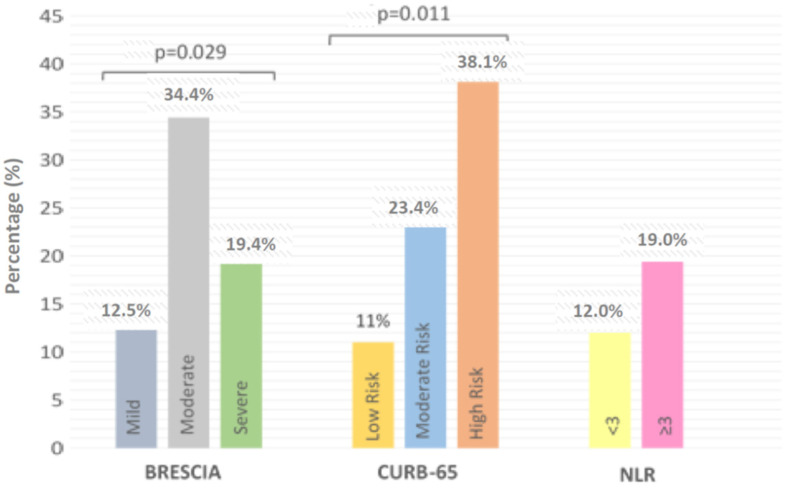
Percentage of patients with positive antiphospholipid antibodies according to severity scale classification of COVID-19.

**Figure 3 viruses-16-01071-f003:**
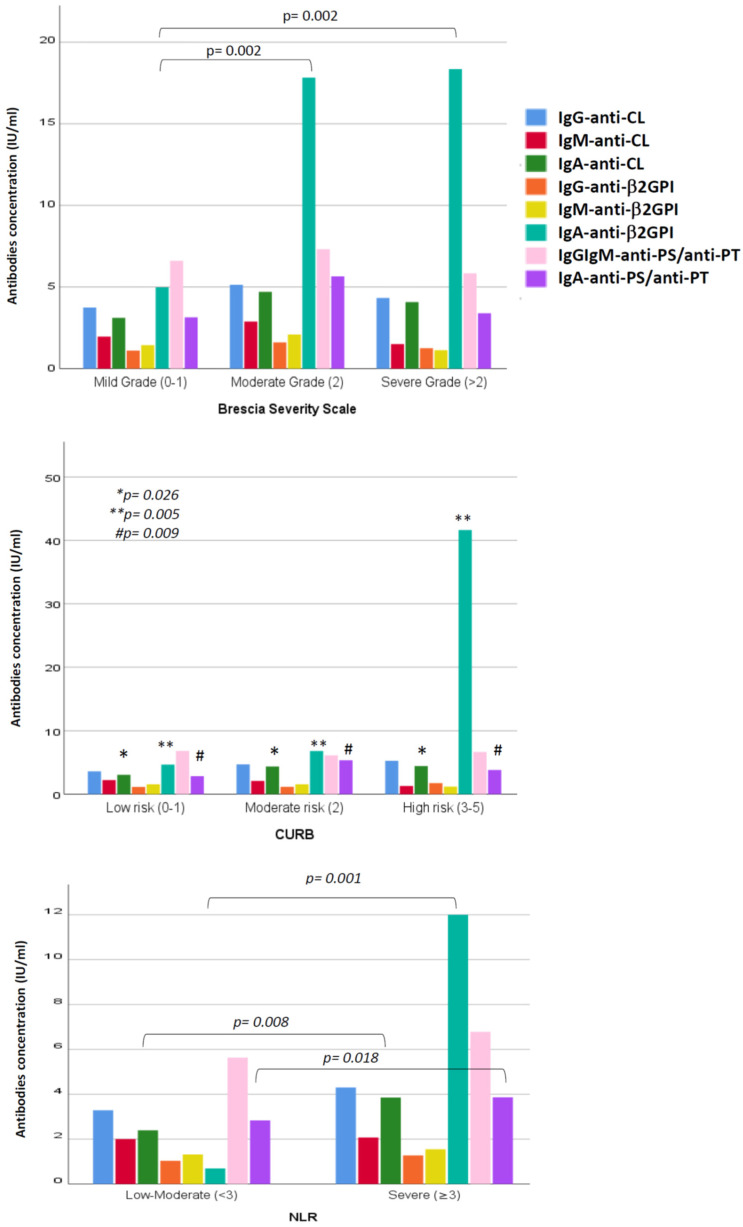
Mean concentration of antiphospholipid antibodies according to severity scale classification of COVID-19.

**Table 1 viruses-16-01071-t001:** Patient characteristics.

General data			
Men	74.2% (n = 118)	Weight (kg)	84.0 ± 15.8
Women	25.8% (n = 41)	Height (cm)	168.8 ± 10.6
Age (years)	63.1 ± 12.3	BMI (kg/cm^2^)	29.4 ± 4.7
Comorbidities			
Hypertension	41.5% (n = 66)	Cardiovascular disease	16.5% (n = 26)
Diabetes	19.5% (n = 31)	Cerebrovascular disease	5.7% (n = 9)
Dyslipidemia	28.9% (n = 46)	Connective tissue diseases	2.5% (n = 4)
Metabolic syndrome	11.3% (n = 18)	Lung disease	18.9% (n = 30)
Thrombophilia	0.6% (n = 1)	Kidney disease	5.0% (n = 8)
Smokers	17.0% (n = 27)	Immunosuppression	8.8% (n = 14)
Neoplasia	15.7% (n = 25)	Pregnant/Puerperal women	0.0% (n = 0)
Treatment prior to admission			
ASA/Clopidogrel	14.5% (n = 23)	Hydroxychloroquine	0.6% (n = 1)
Anticoagulation	8.2% (n = 13)	Contraception	0.0% (n = 0)
Clinical features			
Fever	95.0% (n = 151)	Diarrhoea	22.6% (n = 36)
Cough	74.8% (n = 119)	Anosmia	8.8% (n = 14)
Dyspnea	61.6% (n = 98)	Ageusia	8.8% (n = 14)
Analytical values			
PaO_2_/FiO_2_	191.9 ± 102.2	PT (s)	17.0 ± 9.6
SO_2_/FiO_2_	269.9 ± 121.9	APTT (s)	47.3 ± 24
LDH (U/L)	443.9 ± 207.3	Fibrinogen (mg/dl)	406.4 ± 183.8
Ferritin (ng/mL)	1348.8 ± 1417.7	Lymphocytes (/μL)	936.5 ± 1441.2
D-dimer (μg/mL)	4.1 ± 10.3	IL-6 (pg/mL)	206.6 ± 493.3
Platelets (10^3^/μL)	195.1 ± 99.9	C-Reactive Protein (mg/L)	155.0 ± 79.6
NLR	17.6 ± 58.8		
Percentage of chest affected according to X-ray			
75%	28.9% (n = 46)	50%	52.2% (n = 83)
25%	18.9% (n = 30)		
Thrombotic complications during hospital stay			
DVT	3.1% (n = 5)	Other thrombosis	1.3% (n = 2)
PE	4.4% (n = 7)	TEE in total	7.5% (n = 12)
Treatments during hospital stay			
Corticosteroids	74.8% (n = 119)	Anakinra	1.9 (n = 3)
Corticosteroid boluses	38.4% (n = 61)	LMWH	84.2 (n = 134)
Tocilizumab	39.6% (n = 63)		
Evolution			
Death	6.9% (n = 11)	Readmission	8.8% (n = 14)
ICU	6.9% (n = 11)		

BMI: body mass index. ASA: acetylsalicylic acid. PaO_2_/FiO_2_: arterial oxygen partial pressure (mmHg) to fractional inspired oxygen ratio. SO_2_/FiO_2_: oxygen saturation to fractional inspired oxygen ratio. LDH: lactate dehydrogenase. IL-6: interleukin-6. NLR: neutrophil-to-lymphocyte ratio. DVT: deep vein thrombosis. PE: pulmonary embolism. TEE: thromboembolic events. LMWH: low-molecular-weight heparin.

**Table 2 viruses-16-01071-t002:** Presence and concentrations of antiphospholipid antibodies.

		Presence of aPL		Concentration of aPL (IU/mL)			
**Positive aPL**		**COVID-19**	**Healthy Donors**	**aPL**		**COVID-19**	**Healthy Donors**
anti-CL	*IgG*	3.8% (n = 6)	1.3% (n = 1)	anti-CL	*IgG*	4.1 ± 4.4	2.3 ± 4.7
	*IgM*	2.5% (n = 4)	2.5% (n = 2)		*IgM*	2.1 ± 3.4	2.7 ± 4.2
	*IgA*	0.6% (n = 1)	0.0% (n = 0)		*IgA*	3.6 ± 3.2	3.6 ± 2.1
	*IgG/IgA*	0.6% (n = 1)	0.0% (n = 0)				
	*Total*	7.6% (n = 12)	3.8% (n = 3)				
anti-β2GPI	*IgG*	0.0% (n = 0)	1.3% (n = 1)	anti-β2GPI	*IgG*	1.2 ± 0.9	3.3 ± 4.5
	*IgM*	1.3% (n = 2)	3.8% (n = 3)		*IgM*	1.5 ± 1.9	6.7 ± 20.2
	*IgA*	6.3% (n = 10)	0.0% (n = 0)		*IgA*	10.2 ± 45.3	1.8 ± 1.4
	*Total*	7.6% (n = 12)	5.1% (n = 4)				
anti-PS/anti-PT	*IgG/IgM*	3.8% (n = 6)	12.5% (n = 10)	anti-PS/anti-PT	*IgG/IgM*	6.6 ± 4.4	5.2 ± 6.1
	*IgA*	3.8% (n = 6)	0.0% (n = 0)		*IgA*	3.7 ± 4.2	0.7 ± 0.5
	*IgG/IgM/IgA*	0.6% (n = 1)	0.0% (n = 0)				
	*Total*	8.2% (n = 13)	12.5% (n = 10)				

## Data Availability

The datasets generated during this research or during the data analysis are deposited in the database of the Systemic Autoimmune Diseases Unit of the Department of Internal Medicine, Hospital Universitario Puerta de Hierro Majadahonda and are available from the corresponding author on reasonable request.

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
