# Peer review of "IgA Anti-β2-Glycoprotein I Antibodies as Markers of Thrombosis and Severity in COVID-19 Patients"

_viruses, 2024, doi:10.3390/v16071071_

Round 1
Reviewer 1 Report
Comments and Suggestions for Authors
The authors have investigated the presence of aPL in COVID19. They found a particular higher incidence of IgA-anti-b2GPI in COVID19 patients, with levels associating somewhat to COVID19 severity. The study largely confirms findings from other studies. I have provided an annotated file that highlights the areas for revision. I also have the following suggestions/comments:
1. The authors mention the new classification criteria a few times; I’m not sure why. The classification criteria are for inclusion of subjects into clinical trials; there is no suggestion the subjects discussed in the current manuscript are being considered for entry into a clinical trial? The patients also do not have APS, and no repeat testing was performed. The authors should clarify the reasons for referring to the new classification criteria; it may be better to refer to diagnostic criteria, but also clarify that since repeat testing was not performed, they were not aiming to diagnose APS in these patients.
2. The authors seem to infer that the raised aPL “caused” the TEEs and/or severity of COVID19. There is no evidence presented around cause and effect; the study identifies an interesting observation, and interesting associations, but authors should not infer a causal relationship. D-dimers were also higher in these patients, but this doesn't mean the raised D-dimer caused the increased severity!
3. Throughout the manuscript, authors often use numbers (e.g., %) to 2 decimal points, or mix results with 1 or 2 decimal points; given the study numbers, authors should in general restrict all reported values to a single decimal point. Adding a second decimal point infers an accuracy that is simply not there.
4. Page 2, line 77: “A high…” add references to this statement
5. Page 3, line 108: “percentage of chest X-ray affected”; perhaps better as “percentage of chest affected according to X-ray”
6. Table 1: Why abbreviate a single word ‘Hypertension’ and then define in footer? Just use ‘Hypertension’ in table; define PaFi and SaFI in table footer. “Chest X-ray affected percentage” better as “percentage of chest affected according to X-ray”. A question is posed in the table footer – presumably from one of the authors?
7. Table 2 needs clarification. It is unclear what the data represents. Is this patients only? Or patients and controls, as suggested by the text on page 3, lines 149-150? What do the n values represent? 159, then 158, then 7?
8. Authors should clarify units for the aPL. They use ‘IU/mL’ in methods, then only IU in results. What is the IU used?
9. Sometimes authors revert to comma separator for decimal points; standardise to ‘.’ Separator.
10. What D-dimer assay was used, and are values in FEU or DDU?
11. Figure 1: Right Y-axis; delete % in each number and instead add ‘Percentage (%)’ as a text field for the axis
12. Figure 2 left y-axis. Delete % in each number and instead add ‘Percentage (%)’ as a text field for the axis; remove the decimal points and just use whole numbers. Adding 2 decimal points does not make the data more accurate! Numbers in graph: change ‘,’ separator to ‘.’ And reduce to single decimal point.
13. Figure 3: left y-axis - remove the decimal points and just use whole numbers. Adding 2 decimal points does not make the data more accurate! IU or IU/mL?
14. Discussion: Page 10: line 249: “role in development” – no data is presented to indicate they have a role, just an association.
15. Page 10: lines 297-298: “LA is a very sensitive assay” is it? Sensitive to what?
16. Page 11: lines 301-302: false positives and reason authors did not measure – should be stated earlier in methods
17. Page 12: antiPS and antiPT – standardise – anti-PS and anti-PT used elsewhere

Comments on the Quality of English LanguageEnglish is mostly fine; some minor editing could be done
Author Response
Thank you for the suggestions and comments on our manuscript. The reply and comments to each are shown following the reviewer’s text and the corresponding changes have been included in the manuscript.
The authors have investigated the presence of aPL in COVID19. They found a particular higher incidence of IgA-anti-b2GPI in COVID19 patients, with levels associating somewhat to COVID19 severity. The study largely confirms findings from other studies. I have provided an annotated file that highlights the areas for revision. I also have the following suggestions/comments:
- The authors mention the new classification criteria a few times; I’m not sure why. The classification criteria are for inclusion of subjects into clinical trials; there is no suggestion the subjects discussed in the current manuscript are being considered for entry into a clinical trial? The patients also do not have APS, and no repeat testing was performed. The authors should clarify the reasons for referring to the new classification criteria; it may be better to refer to diagnostic criteria, but also clarify that since repeat testing was not performed, they were not aiming to diagnose APS in these patients.
You are right the classification criteria are used for inclusion of subjects into clinical trials, but sometimes we used them as diagnostic. In our article we want to highlight the antiphospholipid antibodies included in the classification criteria of APS.
- The authors seem to infer that the raised aPL “caused” the TEEs and/or severity of COVID19. There is no evidence presented around cause and effect; the study identifies an interesting observation, and interesting associations, but authors should not infer a causal relationship. D-dimers were also higher in these patients, but this doesn't mean the raised D-dimer caused the increased severity!
Indeed, there is no evidence to infer that the raised aPL caused the TEEs and/or severity of COVID-19 and that is not the conclusion this study wants to convey. So, we have clarified this point with some changes in the aim of the study definition (line 101), in the Discussion (line 283) and in the Conclusions (line 401) sections:
- The aim of this study is to determine in a larger group of patients and healthy donors compared to previous studies, the presence of antiphospholipid antibodies (aPL) in COVID-19 patients and to assess their association with thrombosis and the severity of the disease, and thus whether they can be used as possible patient risk markers and as a guide to their treatment. .
- In this study, the presence of aPL in COVID-19 patients, as well as their association with the development of thrombosis and with the severity of the disease was investigated.
- Further prospective studies are needed with other patient cohorts, increased sample size and long-term follow-up to ascertain the usefulness of these markers in identifying patients with high risk of thrombosis and severe COVID-19.
- Throughout the manuscript, authors often use numbers (e.g., %) to 2 decimal points, or mix results with 1 or 2 decimal points; given the study numbers, authors should in general restrict all reported values to a single decimal point. Adding a second decimal point infers an accuracy that is simply not there.
All the numbers/values reported in the manuscript have been restricted to 1 decimal point.
- Page 2, line 77: “A high…” add references to this statement.
A couple of references has been added to the statement.
- Page 3, line 108: “percentage of chest X-ray affected”; perhaps better as “percentage of chest affected according to X-ray”
The sentence has been changed.
- Table 1: Why abbreviate a single word ‘Hypertension’ and then define in footer? Just use ‘Hypertension’ in table; define PaFi and SaFI in table footer. “Chest X-ray affected percentage” better as “percentage of chest affected according to X-ray”. A question is posed in the table footer – presumably from one of the authors?
Hypertension has been used in the table and removed from the footer. “Percentage of chest affected according to X-ray” sentence has been changed. Definition of PaO2/FiO2 and SO2/FiO2 have been added to the table footer and replaced from the table and Patients and Method section.
BMI: Body Mass Index. ASA: Acetylsalicylic Acid. PaO2/FiO2: arterial oxygen partial pressure (mmHg) to fractional inspired oxygen ratio. SO2/FiO2: oxygen saturation to fractional inspired oxygen ratio. LDH: lactate dehydrogenase. IL6: interleukin-6. NLR: neutrophil-to-lymphocyte ratio. DVT: deep vein thrombosis, PE: pulmonary embolism. TEE: thromboembolic events. LMWH. Low molecular weight heparin.
- Table 2 needs clarification. It is unclear what the data represents. Is this patients only? Or patients and controls, as suggested by the text on page 3, lines 149-150? What do the n values represent? 159, then 158, then 7?
Table 2 has been revised in the paper taking into account the referee´s comments
- Authors should clarify units for the aPL. They use ‘IU/mL’ in methods, then only IU in results. What is the IU used?
The units for the aPL concentration are IU/ml. This point has been corrected in the Results section, in the Table 2 and in the Figure 3.
- Sometimes authors revert to comma separator for decimal points; standardise to ‘.’ Separator.
Comma separators have been replaced by ‘.’ throughout the whole manuscript.
- What D-dimer assay was used, and are values in FEU or DDU?
D dimer has been assayed by Immunoturbidimetry. FEU is the actual measuring units used in mg per mL
- Figure 1: Right Y-axis; delete % in each number and instead add ‘Percentage (%)’ as a text field for the axis
Figure 1 has been modified according to the suggestions.
- Figure 2 left y-axis. Delete % in each number and instead add ‘Percentage (%)’ as a text field for the axis; remove the decimal points and just use whole numbers. Adding 2 decimal points does not make the data more accurate! Numbers in graph: change ‘,’ separator to ‘.’ And reduce to single decimal point.
Figure 2 has been modified according to the suggestions.
- Figure 3: left y-axis - remove the decimal points and just use whole numbers. Adding 2 decimal points does not make the data more accurate! IU or IU/mL?
Figure 3 has been modified according to the suggestions.
- Discussion: Page 10: line 249: “role in development” – no data is presented to indicate they have a role, just an association.
This sentence has been changed. Discussion section, line 283-284.
In this study, the presence of aPL in COVID-19 patients, as well as their association with the development of thrombosis and with the severity of the disease was investigated.
- Page 10: lines 297-298: “LA is a very sensitive assay” is it? Sensitive to what?
This sentence has been modified (line 331).
Lupus anticoagulant assay outcome can be influenced by several factors such as high levels of C-reactive protein and fibrinogen during an acute inflammatory condition which may induce false positive results [15, 30, 31].
- Page 11: lines 301-302: false positives and reason authors did not measure – should be stated earlier in methods
A sentence has been added in Methods section (lines 151-152) and removed from Discussion section
Lupus anticoagulant assay has not been carried out because false positive results can be obtained induced by the acute inflammatory condition of patients.
- Page 12: antiPS and antiPT – standardise – anti-PS and anti-PT used elsewhere
anti-PS and anti-PT are now used throughout the manuscript.

Reviewer 2 Report
Comments and Suggestions for Authors
The topic is interesting and the paper is quite well written. Nevertheless, in my opinion, some parts need to be improved, I have some comments:
1) In this retrospective 19 study, serum samples from 159 COVID-19 patients and 80 healthy donors were analysed for the 20 presence of aPL. Twenty-nine patients (18.24%) and 14 healthy donors (17.5%) were positive for aPL. 21 IgA anti-β2-glycoprotein I antibodies (Anti-β2GPI) was the most frequent type (6.29%) in patients 22 but was not detected in any healthy donor. The positivity of this antibody was found to be signifi23 cantly elevated (p=0.029) in patients with thromboembolic events. Additionally, patients with mod24 erate-severe disease presented a higher aPL positivity than patients with mild disease according to 25 the Brescia (p=0.029) and CURB-65 (p=0.011) severity scales. Moderate and severely ill COVID-19 26 patients also had significantly higher titers of IgA isotype aPL. Please, improve the description of statistically significant data to support the data.
2) In conclusion, COVID-19 patients 27 have a significantly higher positive percentage of IgA isotype aPL than healthy donors. The most 28 frequently detected antibody was IgA anti-β2GPI, which was found to be associated with throm29 bosis and severe COVID-19. Abstract might be beneficial to include a sentence that briefly summarizes the key findings of the study. This can provide readers with a quick overview of the research.
3) . In addition, endothelial cells express ACE2 re45 ceptors allowing infection by SARS-CoV-2. These direct viral effects as well as perivascu46 lar inflammation may contribute to endothelial injury (endothelialitis) [3]. Patients may 47 develop a hypercoagulable state due to this tissue and endothelial injury produced by an 48 unbalanced immune response. 49 Several studies based on autopsies of deceased COVID-19 patients showed a greater 50 degree of endothelialitis, microangiopathy and thrombosis in their lungs, as well as higher 51 tissue expression of IL-6 and TNFα compared to that found in the lungs of patients who 52 died from acute respiratory distress syndrome secondary to influenza A1 (H1N1) infec53 tion and uninfected control lungs [3, 4]. Please, add some information in this paragraph and discuss some references, such as:
a- Radiological-pathological signatures of patients with COVID-19-related pneumomediastinum: is there a role for the Sonic hedgehog and Wnt5a pathways?. ERJ Open Res. 2021;7(3):00346-2021. Published 2021 Aug 23. doi:10.1183/23120541.00346-2021
b- Quantitative Computed Tomography Lung COVID Scores with Laboratory Markers: Utilization to Predict Rapid Progression and Monitor Longitudinal Changes in Patients with Coronavirus 2019 (COVID-19) Pneumonia. Biomedicines. 2024;12(1):120. Published 2024 Jan 6. doi:10.3390/biomedicines12010120
c- Is the Endothelium the Missing Link in the Pathophysiology and Treatment of COVID-19 Complications?. Cardiovasc Drugs Ther. 2022;36(3):547-560. doi:10.1007/s10557-021-07207-w
4) The aim of this study is to determine the presence of antiphospholipid antibodies 81 (aPL) in COVID-19 patients and to assess their clinical association with the development 82 of thrombosis and with the severity of the disease. Please, improve this part and underline the novelty of the study.
5) 2.2. Statistical analysis 127 Qualitative variables are shown as percentages, while quantitative variables are sum128 marized by their mean and standard deviation. The Kolmogorov-Smirnov test was used 129 to evaluate data distribution. For the variables that did not follow a normal distribution, 130 the Wilcoxon, Mann-Whitney U or Kruskal Wallis non-parametric tests were used to com131 pare two continuous variables. The correlation between variables was analyzed using the 132 Spearman test. To compare categorical variables, the χ² test was used, corrected by Yates 133 for expected frequencies <5. A multivariable logistic regression analysis was performed to 134 determine the relationship of antibodies with the severity of the disease. The data were 135 stored in Excel and processed using the statistical program SPSSv25, values being consid136 ered significant when the p value was less than 0.05. Please, underline the statistical tests used to evaluate the data. Please improve the description of statistical tests used to evaluate the data.
6) BMI: Body Mass Index. HTA; Hypertension. ASA: Acetylsalicylic Acid. LDH: lactate dehydrogen144 ase. IL6: interleukin-6. NLR:neutrophil-to-lymphocyte ratio. DVT: deep vein thrombosis, PE: pul145 monary embolism. TEE: thromboembolic events. LMWH. Low molecular weight heparin. DEBERI146 AMOS PONER QUE CONNECTIVE TISSUE DISEASE???? Please pay attention of this typing error.
7) 3. RESULTS. Please, underline the most important results in the text.
8) 4. DISCUSSION 249 In this study, the presence of aPL in COVID-19 patients, as well as their role in the 250 development of thrombosis and relationship with the severity of the disease was investi251 gated. COVID-19 patients were found to have a significantly higher positive percentage 252 of IgA isotype antiphospholipid antibodies than healthy donors. IgA anti-β2GPI was the 253 most frequently detected antibody and was associated with thrombosis and severe 254 COVID-19. The discussion section needs to be improved. It could be interesting to clarify the results obtained and compare them with previous or similar published literature.
9) 5. Conclusions 365 IgA anti-β2GPI was the most frequent aPL detected in the serum of COVID-19 pa366 tients included in this study and was associated with thrombosis and severe COVID-19. 367 Further prospective studies are needed with other patient cohorts, increased sample size 368 and long-term follow-up to identify the contribution of these markers in the pathogenesis 369 of COVID-19. Please, underline the novelty of the study and the possible clinical implications.
Comments on the Quality of English Language
Minor changes of English language are required
Author Response
Reviewer 2
Thank you for the suggestions and comments on our manuscript. The reply and comments to each are shown following the reviewer`s text and the corresponding changes have been included in the manuscript.
The topic is interesting and the paper is quite well written. Nevertheless, in my opinion, some parts need to be improved, I have some comments:
1) In this retrospective 19 study, serum samples from 159 COVID-19 patients and 80 healthy donors were analysed for the 20 presence of aPL. Twenty-nine patients (18.24%) and 14 healthy donors (17.5%) were positive for aPL. 21 IgA anti-β2-glycoprotein I antibodies (Anti-β2GPI) was the most frequent type (6.29%) in patients 22 but was not detected in any healthy donor. The positivity of this antibody was found to be signifi23 cantly elevated (p=0.029) in patients with thromboembolic events. Additionally, patients with mod24 erate-severe disease presented a higher aPL positivity than patients with mild disease according to 25 the Brescia (p=0.029) and CURB-65 (p=0.011) severity scales. Moderate and severely ill COVID-19 26 patients also had significantly higher titers of IgA isotype aPL. Please, improve the description of statistically significant data to support the data.
Description of statistically significant data has been improved in Abstract.
In this retrospective study, serum samples from 159 COVID-19 patients and 80 healthy donors were analysed for the presence of aPL. Twenty-nine patients (18.2%) and 14 healthy donors (17.5%) were positive for aPL. IgA anti-β2-glycoprotein I antibodies (anti-β2GPI) was the most frequent type (6.3%) in patients but was not detected in any healthy donor. The positivity of this antibody was found to be significantly elevated in patients with thromboembolic events (25% vs 5%, p=0.029), in fact patients with positive IgA anti-β2GPI have an incidence of thrombosis over six times higher than those who have normal antibody concentrations [OR (CI 95%) of 6.67 (1.5-30.2), p=0.014]. Additionally, patients with moderate-severe disease presented a higher aPL positivity than patients with mild disease according to the Brescia (p=0.029) and CURB-65 (p=0.011) severity scales. A multivariate analysis showed that positivity for IgA anti-β2GPI is significantly associated with disease severity measured by CURB-65 OR (CI 95%) 17.8 (1.7-187), p=0.0016.
2) In conclusion, COVID-19 patients 27 have a significantly higher positive percentage of IgA isotype aPL than healthy donors. The most 28 frequently detected antibody was IgA anti-β2GPI, which was found to be associated with throm29 bosis and severe COVID-19. Abstract might be beneficial to include a sentence that briefly summarizes the key findings of the study. This can provide readers with a quick overview of the research.
The sentence below summarizing the key findings of the study has been included at the end of Abstract:
IgA anti-β2GPI antibodies were the most frequently detected aPL in COVID-19 patients and were associated with thrombosis and severe COVID-19 and are thus proposed as a possible marker to identify high risk patients.
3) In addition, endothelial cells express ACE2 re45 ceptors allowing infection by SARS-CoV-2. These direct viral effects as well as perivascu46 lar inflammation may contribute to endothelial injury (endothelialitis) [3]. Patients may 47 develop a hypercoagulable state due to this tissue and endothelial injury produced by an 48 unbalanced immune response. 49 Several studies based on autopsies of deceased COVID-19 patients showed a greater 50 degree of endothelialitis, microangiopathy and thrombosis in their lungs, as well as higher 51 tissue expression of IL-6 and TNFα compared to that found in the lungs of patients who 52 died from acute respiratory distress syndrome secondary to influenza A1 (H1N1) infec53 tion and uninfected control lungs [3, 4]. Please, add some information in this paragraph and discuss some references, such as:
a- Radiological-pathological signatures of patients with COVID-19-related pneumomediastinum: is there a role for the Sonic hedgehog and Wnt5a pathways?. ERJ Open Res. 2021;7(3):00346-2021. Published 2021 Aug 23. doi:10.1183/23120541.00346-2021.
b- Quantitative Computed Tomography Lung COVID Scores with Laboratory Markers: Utilization to Predict Rapid Progression and Monitor Longitudinal Changes in Patients with Coronavirus 2019 (COVID-19) Pneumonia. Biomedicines. 2024;12(1):120. Published 2024 Jan 6. doi:10.3390/biomedicines12010120
c- Is the Endothelium the Missing Link in the Pathophysiology and Treatment of COVID-19 Complications?. Cardiovasc Drugs Ther. 2022;36(3):547-560. doi:10.1007/s10557-021-07207-w
The following paragraphs have been included in the Introduction section, discussing the suggested references:
The evidence from many COVID-19 studies points to endothelial damage as a key component in the progression of the disease to its later complicated stages. Endothelial damage is associated with the loss of the anticoagulant properties of the endothelium, which may contribute to the hypercoagulable state of these patients as well as an overactivation of the complement cascade in SARS-CoV-2 which in turn can also promote acute and chronic inflammation, intravascular coagulation and endothelial cell injury [5]. The endothelial damage caused by COVID-19 is therefore at the crossroad of the hypercoagulable state, impaired fibrinolysis, activation of the complement system and the degradation of the glycocalyx layer, all of which are processes linked in the pathogenesis of COVID-19 complications.
The identification of the risk of progression to severe disease is crucial in preventing respiratory failure and mortality in COVID-19 patients. Elevated C-reactive protein. D-dimer, lactate dehydrogenase and interleukin-6 as well as severe lymphopenia have been reported to be associated significantly with worse outcomes [11]. Recently, Kang DH et al have explored the prognostic value of quantitative high-resolution computed tomography (QCT) lung COVID scores along with laboratory inflammation markers and found that patients with a high mixed disease pattern score (≥10%) were likely to experience rapid progression within seven days, suggesting that QCT COVID scores at admission could predict rapid progression in COVID-19 patients [12].
4) The aim of this study is to determine the presence of antiphospholipid antibodies 81 (aPL) in COVID-19 patients and to assess their clinical association with the development 82 of thrombosis and with the severity of the disease. Please, improve this part and underline the novelty of the stu
The following sentence has been added to the text (lines 101-105).
The aim of this study is to determine in a larger group of patients and healthy donors compared to previous studies, the presence of antiphospholipid antibodies (aPL) in COVID-19 patients and to assess their association with thrombosis and the severity of the disease, and thus whether they can be used as possible patient risk markers and as a guide to their treatment. .
5) 2.2. Statistical analysis 127 Qualitative variables are shown as percentages, while quantitative variables are sum128 marized by their mean and standard deviation. The Kolmogorov-Smirnov test was used 129 to evaluate data distribution. For the variables that did not follow a normal distribution, 130 the Wilcoxon, Mann-Whitney U or Kruskal Wallis non-parametric tests were used to com131 pare two continuous variables. The correlation between variables was analyzed using the 132 Spearman test. To compare categorical variables, the χ² test was used, corrected by Yates 133 for expected frequencies <5. A multivariable logistic regression analysis was performed to 134 determine the relationship of antibodies with the severity of the disease. The data were 135 stored in Excel and processed using the statistical program SPSSv25, values being consid136 ered significant when the p value was less than 0.05. Please, underline the statistical tests used to evaluate the data. Please improve the description of statistical tests used to evaluate
Statistical analysis description has been improved.
Categorical variables were reported as frequencies and percentages while continuous variables were presented by their mean and standard deviation. The Kolmogorov-Smirnov test was used to evaluate data distribution. The correlation between the aPL concentrations and clinical variables was analyzed using the Spearman test. The significance of the differences and outcomes about the presence of antiphospholipid antibodies, thrombosis and disease severity between aPL positive patients and patients with non-positive aPL were determined by the Chi-square, Fisher’s or Student’s t-test, as appropriate. A multivariable logistic regression analysis was performed to determine the association of antibodies with the severity of the disease. The data were stored in Excel and processed using the statistical program SPSSv25. For all the analyses, a significance level of 0.05 was set.
6) BMI: Body Mass Index. HTA; Hypertension. ASA: Acetylsalicylic Acid. LDH: lactate dehydrogen144 ase. IL6: interleukin-6. NLR:neutrophil-to-lymphocyte ratio. DVT: deep vein thrombosis, PE: pul145 monary embolism. TEE: thromboembolic events. LMWH. Low molecular weight heparin. DEBERI146 AMOS PONER QUE CONNECTIVE TISSUE DISEASE???? Please pay attention of this typing error.
The Table 2 footer has been modified and a typing error has been removed.
7) 3. RESULTS. Please, underline the most important results in the text.
We are not sure about the possibility of underlining the text in the manuscript. Anyway, the most important results are described in the Results section and are summarized as follows:
3.1. Presence of antiphospholipid antibodies (aPL)
(Lines 185-186). Interestingly, IgA anti-β2GPI was the most frequently detected in 10 patients (6.3%) but was not detected in any healthy donor.
3.2. Antiphospholipid antibodies, thrombosis and coagulation test in patients with COVID-19
(Lines 218-223). Analysis of the relationship between the presence of aPL and the incidence of thrombosis showed an OR (CI 95%) of 6.67 (1.5-30.2), p=0.014, meaning that patients with positive IgA anti-β2GPI have an incidence of thrombosis over six times higher than those who have normal antibody concentrations; the patients with thrombosis had a significantly higher positive percentage of IgA anti-β2GPI than the group of patients without thrombosis (25.0% vs 4.8%, p=0.029) (Figure 1).
3.3. Antiphopholipid antibodies and severity of COVID-19
(Lines 248-252). Considering each aPL subtype concentrations, significant differences (p =0.002) were only found for IgA anti-β2GPI, the higher the concentration of this antibody the greater the severity of the disease (4.9 ± 29.7 IU/ml for mild grade, 17.8 ± 60.0 IU/ml for moderate grade and 18.4 ± 63.8 IU/ml for severe grade) (Figure 3).
(Lines 267-268). A multivariate analysis showed that positivity for IgA anti-β2GPI is significantly associated with disease severity measured by CURB-65 OR (CI 95%) 17.8 (1.7-187), p=0.0016.
8) 4. DISCUSSION 249 In this study, the presence of aPL in COVID-19 patients, as well as their role in the 250 development of thrombosis and relationship with the severity of the disease was investi251 gated. COVID-19 patients were found to have a significantly higher positive percentage 252 of IgA isotype antiphospholipid antibodies than healthy donors. IgA anti-β2GPI was the 253 most frequently detected antibody and was associated with thrombosis and severe 254 COVID-19. The discussion section needs to be improved. It could be interesting to clarify the results obtained and compare them with previous or similar published literature.
In the discussion, we compared our results with similar published literature
(Lines 288-291) A similar prevalence of aPL was found in our COVID-19 patients (18.2%) compared to our healthy donor group (17.5%). In other studies, the prevalence of aPL with COVID-19 ranged from 1% to 95% [13-19]. Taha et al. conducted a meta-analysis and found that a pooled prevalence rate of one or more aPL was 46.8% (95% CI 36.1% to 57.8%) [20].
(Lines 302-306) Although a widespread presence of aPL has been reported in patients with COVID-19, discrepancies in the data on the degree of prevalence and how it affects the pathogenesis of thrombosis exists. This may be a result of the diversity of the study population, the disease severity, the point in time during the disease when aPL were determined, the aPL type measured and the detection procedure.
(Lines 308-313) IgA anti-β2GPI was the most frequently detected antibody in the COVID-19 patients of our study (6.3%) and presented with moderate and high titres. This was also observed by Xiao et al. [24], Melayah et al [25] and Serrano et al [26] who reported IgA anti-β2GPI as the most frequently occurring aPL in COVID-19 patients (28.8%, 16.9% and 14.9% respectively). This finding suggests that COVID-19 induced IgA isotype aPL. Other authors reported an IgA anti-β2GPI prevalence of between 6.6% and 12% in critically ill patients [14, 15, 17]
IgA anti-β2GPI was associated with thrombosis and severe COVID-19.
(Lines357-386) In the literature, there are divergent results on the presence of aPL and the risk of thrombotic events in COVID-19 patients. Xiao et al. found that patients with multiple aPL including IgA and IgG anti-β2GPI and IgA and IgG anti-CL displayed a significantly higher incidence of cerebral infarction compared to patients who were negative for aPL (p=0.023) [24], while Amezcua-Guerra et al. suggested an association with pulmonary thromboembolism [29]. Le Joncour et al. showed a meaningful association of anticardiolipin and IgA anti–β2GPI with the occurrence of thrombotic events [36] and Gatto et al. found that patients with at least one aPL positive test show a trend in developing thrombosis compared with patients who were aPL negative [13].
However, other published studies seem to rule out the relationship between aPL and thrombosis in COVID-19 patients. This may be because of the low number of patients included in each study; nevertheless, our study is one of the studies with the highest number of patients. In addition, most of these studies carried out lupus anticoagulant assays and only a few of them analysed IgA anti-β2GPI; it should also be borne in mind that all these patients, as well as those included in our study, received prophylactic treatment with LMWH, which probably results in a lower occurrence of TEE [13, 15-17, 20].
As the pathogenesis of COVID-19 is related to an exaggerated inflammatory and immunological response, a higher concentration of aPL with disease severity might thus be expected. In our study, the results showed that the more critical patients had greater positivity for any type of aPL as well as a significant increase of IgA isotype aPL concentrations. This was also found by Hasan Ali et al. who compared two cohorts with mild and severe COVID-19 and observed significantly elevated IgA anti-CL and IgA anti-β2GPI in the severe COVID-19 cohort [22] and by Garcia-Arellano et al. who found a significantly higher prevalence of IgA anti-β2GPI in moderate to severe COVID-19 patients [37]. In other published studies, those that registered the highest prevalence of aPL were those that selected patients with severe COVID-19 or in patients admitted to the ICU [15, 16, 20, 24, 38]. This can be partially explained by the extensive inflammation, cellular damage and apoptosis in critically ill patients that can induce aPL production. These findings suggest that the presence of aPL in COVID 19 patients, especially IgA isotype, could be used as markers for COVID-19 severity.
9) 5. Conclusions 365 IgA anti-β2GPI was the most frequent aPL detected in the serum of COVID-19 pa366 tients included in this study and was associated with thrombosis and severe COVID-19. 367 Further prospective studies are needed with other patient cohorts, increased sample size 368 and long-term follow-up to identify the contribution of these markers in the pathogenesis 369 of COVID-19. Please, underline the novelty of the study and the possible clinical implications.
Possible clinical implications are now included in the conclusions-see below
IgA anti-β2GPI was the most frequent aPL detected in the serum of COVID-19 patients included in this study and was associated with thrombosis and severe COVID-19. Further prospective studies are needed with other patient cohorts, increased sample size and long-term follow-up to ascertain the usefulness of these markers in identifying patients with high risk of thrombosis and severe COVID-19.

Round 2
Reviewer 2 Report
Comments and Suggestions for Authors
The manuscript was partially improved, no further comments. I suggest to underline in the discussion the novelthy of this work
Comments on the Quality of English LanguageMinor changes of English language are required
Author Response
Reviewer 2
Comments and Suggestions for Authors
Thank you for the suggestions and comments on our manuscript. The reply and comments to each are shown following the reviewer`s text and the corresponding changes have been included in the manuscript.
1.-The manuscript was partially improved, no further comments. I suggest to underline in the discussion the novelthy of this work
To underline the novelty of this work we suggest that the following is added at the beginning of the discussion at lines 285-292:
Our novel findings are that COVID-19 patients have a significantly higher positive percentage of IgA isotype antiphospholipid antibodies than healthy donors. As IgA anti-β2GPI was the most frequently detected antibody and was associated with thrombosis and severe COVID-19 it is proposed that this should be used as a marker for disease severity. Thus, it is suggested that Covid-19 patients should be screened for IgA anti-β2GPI and those found to have higher titers subject to close follow up to prevent the development of thrombosis.
2.- In the Results section the following sentence has been modified after reference to Table 2 to clarify the differences obtained between patients and healthy donors.
Interestingly, nineteen COVID-19 patients (12%) presented a positive percentage of IgA isotype aPL, but no healthy donor had IgA isotypes. IgA anti-β2GPI was the most frequently detected, being present in 10 (6.3%) of the COVID-19 patients.
3.- Comments on the Quality of English Language: Minor changes of English language are required
The manuscript has been reviewed and some changes have been made at lines 24, 92, 100, 151 - 152, 158, 170, 187, 231, 284, 321.
4.- Following the editor´s recommendation, we have also double checked that all the literature is correctly cited.